# Frustration-driven $C_4$ symmetric order in a naturally-heterostructured superconductor $Sr_2VO_3FeAs$

Jong Mok Ok[1,2], S.-H. Baek [3], C. Hoch[4], R.K. Kremer[4], S.Y. Park [5], Sungdae Ji[1,5], B. Büchner[3], J.-H. Park[1,5,6], S.I. Hyun[7], J.H. Shim[7], Yunkyu Bang[8], E.G. Moon[9], I.I. Mazin[10] & Jun Sung Kim[1,2]

A subtle balance between competing interactions in iron-based superconductors (FeSCs) can be tipped by additional interfacial interactions in a heterostructure, often inducing exotic phases with unprecedented properties. Particularly when the proximity-coupled layer is magnetically active, rich phase diagrams are expected in FeSCs, but this has not been explored yet. Here, using high-accuracy $^{75}As$ and $^{51}V$ nuclear magnetic resonance measurements, we investigate an electronic phase that emerges in the FeAs layer below $T_0 \sim 155$ K of $Sr_2VO_3FeAs$, a naturally assembled heterostructure of an FeSC and a Mott-insulating vanadium oxide. We find that frustration of the otherwise dominant Fe stripe and V Neel fluctuations via interfacial coupling induces a charge/orbital order in the FeAs layers, without either static magnetism or broken $C_4$ symmetry, while suppressing the Neel antiferromagnetism in the $SrVO_3$ layers. These findings demonstrate that the magnetic proximity coupling stabilizes a hidden order in FeSCs, which may also apply to other strongly correlated heterostructures.

[1] Department of Physics, Pohang University of Science and Technology, Pohang 790-784, Korea. [2] Center for Artificial Low Dimensional Electronic Systems Institute for Basic Science, Pohang 790-784, Korea. [3] IFW Dresden, Helmholtzstr. 20, 01069 Dresden, Germany. [4] Max-Planck-Institut für Festkörperforschung, Heisenbergstraβe 1, D-70569 Stuttgart, Germany. [5] Max Planck POSTECH Center for Complex Phase Materials Pohang University of Science and Technology, Pohang 790-784, Korea. [6] Division of Advanced Materials Science Pohang University of Science and Technology, Pohang 790-784, Korea. [7] Department of Chemistry, Pohang University of Science and Technology, Pohang 790-784, Korea. [8] Department of Physics, Chonnam National University, Gwangju 500-757, Korea. [9] Department of Physics, Korea Advanced Institute of Science and Technology, Daejeon 305-701, Korea. [10] Naval Research Laboratory code 6390, 4555 Overlook Avenue S.W., Washington, DC 20375, USA. Correspondence and requests for materials should be addressed to S.-H.B. (email: sbaek.fu@gmail.com) or to J.S.K. (email: js.kim@postech.ac.kr)

In strongly correlated electron materials, including cuprates, transition metal oxides (TMOs), and iron-based superconductors (FeSCs), competing interactions of spin, charge, and orbital degrees of freedom lead to complex and rich phase diagrams, extremely sensitive to external perturbations. Especially impressive is the modification of the phase diagram via introducing interfacial interactions, as intensively studied for the heterostructures of high-$T_c$ cuprates[1–5] or TMOs[6,7], showing the enhanced $T_c$ or emergent phases that cannot be stabilized in their constituent layer alone. The similar effect has also been found in FeSCs, for example, in FeSe monolayers on top of nonmagnetic $SrTiO_3$[8–10] showing drastically enhanced $T_c$, arguably higher than 100 K. Although the underlying mechanism is yet to be confirmed, the interfacial coupling is considered to be critical and may further enhance $T_c$ in the superlattice[11]. Of particular interest is when the proximity-coupled layer is strongly correlated and magnetically active. As found in heterostructures of high-$T_c$ cuprates and magnetic TMOs[3–5], additional interfacial spin interaction, e.g., in proximity of a Mott insulator, may also induce distinct ground states in FeSCs[12], which however has not been explored so far.

$Sr_2VO_3FeAs$ is an unusual naturally assembled superlattice of $[SrFeAs]^{+1}$ and $[SrVO_3]^{-1}$ layers[13]. Initially $Sr_2VO_3FeAs$ was thought to have, because of the V bands, an unusual Fermi surface topology, incompatible with $s^{\pm}$ superconductivity scenario driven by spin fluctuation[14]. However, it was soon realized that the V $3d^2$ electrons in the $SrVO_3$ layer are strongly correlated and form a Mott-insulating state[15–17], while the partially filled Fe $3d^6$ state in the FeAs layer has considerable itinerancy and superconducts at $T_c \sim 35$ K[13,14,18]. These contrasting ground states in $Sr_2VO_3FeAs$ make this system prototypical for strongly correlated heterostructures based on FeSCs and TMOs. $Sr_2VO_3FeAs$ has the Fermi surface structure similar to that in other FeSCs[15,17], and thus is expected to show either the stripe antiferromagnetic (AFM) order with the wave vector $\mathbf{Q} = (\pi, 0)$, or the corresponding nematic phase, or enhanced spin fluctuations at low temperature with the same wave vector[19]. There is in fact a second-order transition observed at $T_0 \sim 155$ K with a sizable entropy loss of $\sim 0.2R\ln 2$ ($R$ is the gas constant)[20–22]. With no evidence of a static magnetic order or another apparent symmetry breaking, the hidden nature of this phase transition, similar to the famous hidden order in underdoped cuprates or a heavy fermion system $URu_2Si_2$, remains elusive and controversial[20–26], posing a challenge to our understanding of the physics of FeSCs in proximity of a Mott insulator.

Here we report that an emergent electronic phase is developed below $T_0 = 155$ K in $Sr_2VO_3FeAs$, which is highly distinct in nature from the transitions found in other FeSCs. Using high-accuracy $^{75}As$ and $^{51}V$ nuclear magnetic resonance (NMR) measurements on single crystals under various field orientations, we unambiguously show that the transition occurs in the FeAs layer, not the $SrVO_3$ layer, without breaking either time reversal symmetry or the underlying tetragonal lattice symmetry. This implies that the typical stripe AFM and $C_2$ nematic phases in the FeAs layers as well as the Neel antiferromagnetism in the $SrVO_3$ layer are significantly suppressed by the interfacial coupling between itinerant iron electrons and localized vanadium spins. We propose that the observed phase is a $C_4$-symmetric charge/ orbital order, which to our knowledge has never been observed in iron or vanadium-based materials, triggered by frustration of the otherwise dominant Fe stripe and V Neel fluctuations. Such a strong modification of the ground state is not common in other strongly correlated TMO heterostructures[3–5], which highlights that FeSCs, itinerant systems with complex interplay of spin/ charge/orbital degrees of freedom, have competing ground states related to the Fermi surface instabilities, and thus are extremely sensitive to additional interfacial interactions in heterostructures.

## Results

**Transport and magnetic properties.** Our transport and magnetic measurements on single crystalline $Sr_2VO_3FeAs$ shown in Fig. 1b, c confirm that the transition at $T_0$ is intrinsic. A weak, but discernible, anomaly is observed at $T_0 \sim 155$ K in the resistivity ($\rho$), even more pronounced in its temperature derivative $d\rho/dT$. The magnetic susceptibility $\chi(T)$ also shows an anomaly at $T_0$. Above $T_0$, $\chi(T)$ is several times larger than in typical FeSCs and follows the Curie–Weiss law with a Curie–Weiss temperature $T_{CW} \sim -100$ K (see Supplementary Note 2). The effective magnetic moment is consistent with $S = 1$ expected for the $V^{3+}$ ions (Fig. 1a), suggesting that $\chi(T)$ is dominated by localized V spins. At $T_0 \sim 155$ K, $\chi(T)$ for both $H \parallel ab$ and $H \parallel c$ exhibits a small jump, which corresponds to a magnetization of $\sim 10^{-3}$ $\mu_B$/f.u., three orders of magnitude smaller than typical values of $V^{3+}$ ions ($\sim 1.8$ $\mu_B$) in vanadium oxides[16] and Fe ions ($\sim 0.8$ $\mu_B$) in FeSCs[27]. Such weak anomalies in $\rho(T)$ and $\chi(T)$, in contrast to a strong one in the specific heat[20–22], question the previous conjectures of a long-range ordering of either V or Fe spins[20–26], and suggest that this weak ferromagnetic response is only a side effect of the true transition. However, another anomaly at $T_N \sim 45$ K in both $\chi_{ab}(T)$ and $\chi_c(T)$ turns out to reflect a long-range ordering of Fe, but still not V spins, as discussed below. Notably, neither transition is consistent with the typical stripe AFM or nematic orders for FeSCs.

**$^{75}As$ and $^{51}V$ NMR spectroscopy.** To gain further insight into the transition at $T_0$ on a microscopic level, we measured NMR on $^{75}As$ and $^{51}V$ nuclei as a function of temperature for field orientations parallel to $a$ (100), $c$ (001), and the (110) directions (Fig. 2 and the Supplementary Fig. 4). The $^{51}V$ probes the V spin order directly and the $^{75}As$ is a proxy for the Fe sites, which allows us to probe the two magnetic ions separately. A dramatic change of the $^{75}As$ line occurs near $T_0 \sim 155$ K as shown in Fig. 2a, consistent with the anomalies in $\rho(T)$ and $\chi(T)$. Near 180 K, the $^{75}As$ signal starts to lose its intensity rapidly and is not detectable between 150 and 170 K due to the shortening of the spin–spin relaxation time $T_2$ (ref. [26]). Strikingly, the signal recovers below $\sim 150$ K at substantially higher frequencies, in a similar fashion for both field orientations. We emphasize that these behaviors of $^{75}As$ NMR have never been observed so far in other FeSCs as clearly shown in Supplementary Fig. 5. This is better shown in terms of the Knight shift $^{75}\mathcal{K} \equiv (f - \nu_0)/\nu_0$, where $\nu_0 \equiv \gamma_n H$ with the nuclear gyromagnetic ratio $\gamma_n$ (see Fig. 2c). $^{75}\mathcal{K}$ changes abruptly at $T_0 \sim 155$ K without any peak splitting or broadening of the full-width at half-maximum (FWHM) across $T_0$. Conversely, the $^{51}V$ line barely shifts below $T_0$ and down to 20 K (Fig. 2b, d), while its FWHM gradually increases below $T_0$. The nearly unchanged $^{51}V$ NMR line signals that the V spins remain disordered down to low temperatures. This contrasting behavior of the $^{75}As$ and $^{51}V$ spectra unambiguously proves that the transition at $T_0$ occurs in the FeAs layer and not in the $SrVO_3$ layer, contrary to previous claims[20–25].

Having established that the phase transition at $T_0$ occurs in the FeAs layer, we examined the low-energy Fe spin dynamics, as probed by the $^{75}As$ spin–lattice relaxation rate $T_1^{-1}$, which reflects local spin fluctuations. As shown in Fig. 3, at $T \gtrsim 240$ K, $(T_1T)^{-1}$ exhibits a typical Curie–Weiss-like behavior with an anisotropy $T_{1,a}^{-1}/T_{1,c}^{-1} \approx 1.5$. This is expected for a directionally disordered state with local stripe AFM correlations with $\mathbf{Q} = (\pi, 0)$ and has been observed in many FeSCs[28,29] (see Supplementary Note 6). With lowering temperature, a critical slowdown of the $(\pi, 0)$ spin fluctuations usually condenses into the $C_2$ stripe AFM phase. For $Sr_2VO_3FeAs$, however, this critical growth is arrested at $T \sim 200$ K, showing a broad peak of $(T_1T)^{-1}$ with an unusually large

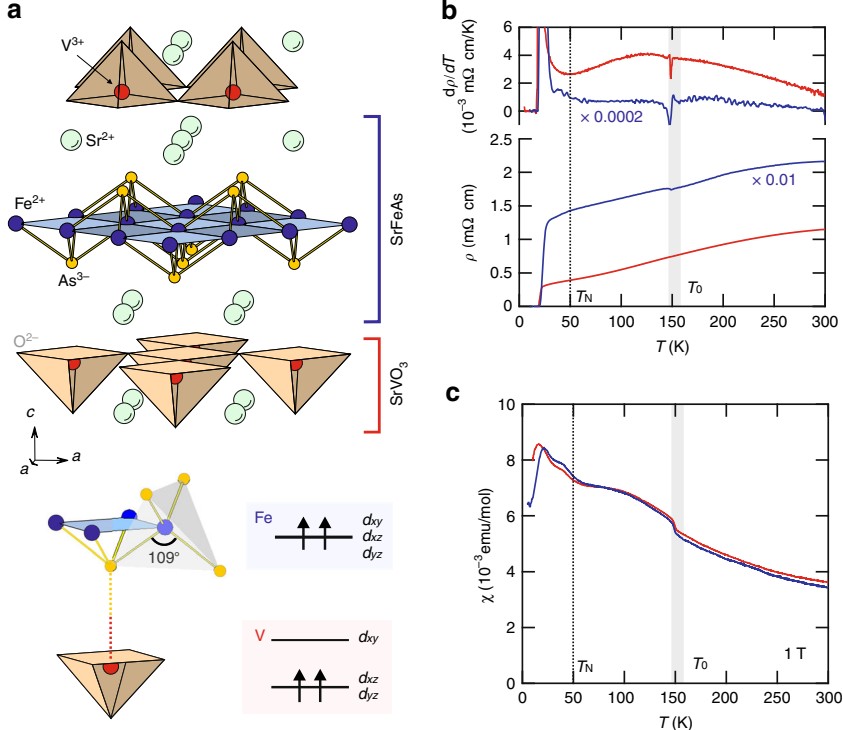

**Fig. 1** Basic properties of $Sr_2VO_3FeAs$. **a** The crystal structure of $Sr_2VO_3FeAs$ as a naturally assembled heterostructure of the $[SrFeAs]^{+1}$ and $[SrVO_3]^{-1}$ layers. V ions form a network of corner-sharing tetrahedra, while FeAs layers consist of edge-sharing $FeAs_4$ tetrahedra as in other iron-based superconductors. The local structure of an $FeAs_4$ tetrahedron and a $VO_3$ pyramid is highlighted at the bottom. $FeAs_4$ tetrahedra provide a moderate cubic crystal field splitting, much smaller than the band widths, while the $VO_3$ unit is missing one O entirely, and thus develops a strong Jahn–Teller splitting. The $d_{xy}$ orbital is pushed up, and the two V $d$ electrons occupy the $d_{xz}$ and $d_{yz}$ states, forming an $S = 1$ spin. The Fe and V planes are bridged by the As atoms, as indicated by the dashed line. **b** The resistivity $\rho(T)$ in the $ab$ plane (red) and along the $c$ axis (blue) shows a weak anomaly at $T_0 \sim 155$ K (blue highlighted region), which is more clear in their temperature derivatives (top panel). The large $c$-axis resistivity, which was scaled down by a factor of 100, is consistent with the quasi-2D nature of the material. **c** The magnetic susceptibility $\chi(T)$, taken at $H = 1$ T for $H \perp c$ and $H \parallel c$, shows clear anomalies at $T_0$ and also at $T_N$ $\sim 45$ K

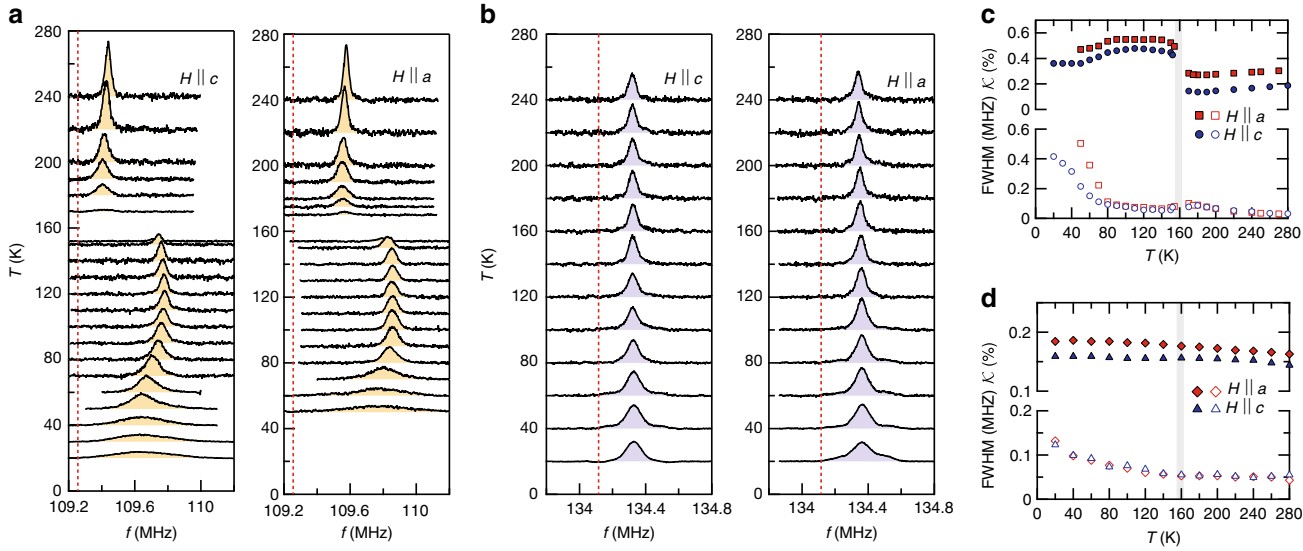

**Fig. 2** NMR spectra and their analysis for the $Sr_2VO_3FeAs$ single crystal. $^{75}As$ (**a**) and $^{51}V$ (**b**) NMR spectra as a function of temperature, measured at $H = 15$ and 12 T, respectively, for the fields oriented along the $a$ and $c$ axes. The unshifted Larmor frequency ($\nu_0 \equiv \gamma_n H$) is marked by the red vertical lines. While the $^{51}V$ spectrum is nearly temperature independent down to 20 K, the $^{75}As$ spectra in both field directions show a sudden shift at $T_0 \sim 155$ K. **c**, **d** Temperature dependences of the $^{75}As$ and $^{51}V$ spectra in terms of the Knight shift ($\mathcal{K}$) and the full-width at half-maximum (FWHM), respectively. Below $T_0$, a nearly isotropic large jump of the $^{75}As$ Knight shift takes place without any magnetic line broadening, contrasting with the $^{51}V$ spectra that remain unchanged

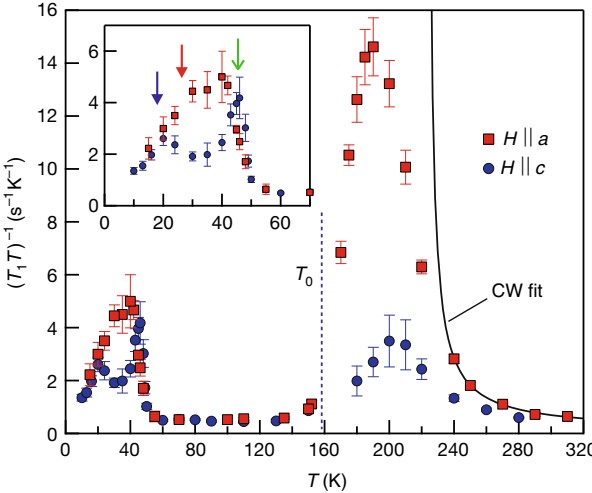

**Fig. 3** Fe spin fluctuations. Temperature dependence of the $^{75}$As spin–lattice relaxation rate divided by temperature $(T_1T)^{-1}$ measured at 15 T. The error bars reflect the uncertainty in the fitting procedure. At high temperatures, $(T_1T)^{-1}$ is well described by a Curie–Weiss (CW) law (solid line). Below ~240 K, it deviates from the diverging behavior and drops at lower temperatures, forming a large peak centered at ~190 K. At low temperatures below 120 K, $(T_1T)^{-1}$ reaches a constant value comparable to that observed at the high temperature limit, implying that the spin fluctuations are completely gapped out. $(T_1T)^{-1}$ sharply turns up at ~50 K, indicating critical slowing down of spin fluctuations toward a magnetic order. In the inset, $T_N \sim 45$ K (green arrow) was identified from the sharp peak observed for $H \parallel c$. $(T_1T)^{-1}$ drops at $T_c$, determined by the resistivity measurements under $H \parallel ab$ (red arrow) and $H \parallel c$ (blue arrow), microscopically probing bulk superconductivity in the magnetically ordered state

$T_{1,a}^{-1}/T_{1,c}^{-1} \approx 6$, and then the fluctuations harden all the way down to $T_0$. Across $T_0$, $(T_1T)^{-1}$ barely changes and then quickly reaches a constant below $T_0$, behaving as a paramagnetic metal. This completely unexpected behavior in both $^{75}\mathcal{K}$ and $(T_1T)^{-1}$ confirms that the transition at $T_0$ in Sr$_2$VO$_3$FeAs is unlike any transitions observed in FeSCs so far.

Let us now discuss possible orders established below $T_0$. First of all, we can eliminate the usual suspects: stripe, double-Q[30–33], and bicollinear[34] AFM orders, observed in other FeSCs. In the first case $Q = (\pi, 0)$ and the Fe spins aligned along the $a$ axis ($s \parallel a$) generate a hyperfine field $H_{hf} \sim 1.5$ T along the $c$ axis. This would be visible in the $^{75}$As NMR spectra as a peak splitting of ~10 MHz for $H \parallel c$, which is far larger than the FWHM of our spectra (~0.05 MHz) and easily detectable. Similarly, for $s \parallel c$, an $^{75}$As peak splitting is expected for $H \parallel a$. Even for $s \parallel b$, in which case no transferred $H_{hf}$ and thus no peak splitting are expected, considerable line broadening due to the directional fluctuations of Fe spins should be seen in experiments. Neither splitting nor broadening is observed in our experiments (Fig. 2a). For the double-Q AFM state[30–33], a combination of two spin density waves with $Q = (\pi, 0)$ and $(0, \pi)$, the magnetization vanishes at one of the two Fe sublattices and is staggered in the other. Thus, the $^{75}$As peak splitting is expected for either $H \parallel c$ ($s \parallel a$) or $H \parallel a$ ($s \parallel c$), as discussed in the Supplementary Note 5, which can be ruled out by experiments. The bicollinear AFM order[34] can also be excluded with even more confidence. In this case, the As environment is spin-imbalanced (three neighboring Fe spins are aligned in one direction, and the fourth one in the opposite), and already a plain exchange coupling would generate two inequivalent As sites and thus a measurable splitting for any direction of external fields. Similarly, other AFM orders with more

complicated spin structures, such as a plaquette AFM order, are excluded as discussed in the Supplementary Note 5. This conclusion is further supported by the absence of the diverging behavior in $(T_1T)^{-1}$ across $T_0$ (Fig. 3).

Having excluded static magnetic order, we consider now nematic or, as it is occasionally called, vestigial partners of various AFM orders. The only nematic order observed so far in FeSCs is the stripe-nematic order that creates an imbalance in the orbital population between Fe $d_{xz}$ and $d_{yz}$ states (Fig. 4b). This, in turn, induces an imbalance between As $p_x$ and $p_y$ orbitals and dipolar in-plane anisotropy of the As Knight shift in the twinned crystals, as observed in e.g., LaFeAsO for $H \parallel a$ below the nematic transition temperature[29]. A similar behavior is expected for the nematic partner of the bicollinear order (Fig. 4c), which breaks the $C_4$ symmetry such that the (110) and (1$\bar{1}$0) directions are not equivalent[35,36]. If the generated imbalance between the corresponding orbital Fe-$d_{xz} \pm d_{yz}$ is of the same order as in the stripe-nematic case, a peak splitting for $H \parallel$ (110) should be detected. And, for the nematic partner of the plaquette magnetic order, two inequivalent sites and thus a sizable splitting are expected for every field direction. Yet, none of these signatures appear in our $^{75}$As spectra for $H \parallel a$ (100), $c$ (001), and (110) directions (Fig. 2a, b and the Supplementary Fig. 4). Furthermore, our single-crystal X-ray diffraction (XRD) (see Supplementary Note 1) as well as the recent ARPES study[17] do not reveal any signature of a $C_4$ symmetry breaking.

Since the transition at $T_0$ retains the $C_4$ symmetry, and in the absence of a long-range magnetic order, this transition must generate a change in the relative occupations of the $C_4$ orbitals, namely $d_{xy}$, $d_{z^2}$, $d_{x^2-y^2}$, and $d_{xz} \pm id_{yz}$. Given that at high temperature we see clear indications of strong spin fluctuations, we looked for a spin-driven scenario conserving the $C_4$ symmetry; a good candidate is the vestigial (nematic) partner of the double-Q AFM order[37]. It can be visualized (Fig. 4d) as a superposition of two charge/orbital density waves with $Q = (\pi, 0)$ and $(0, \pi)$, which preserves the $C_4$ symmetry without unit-cell doubling. This phase has a broken translational symmetry in the Fe-only square lattice, but not in the unit cell doubled to include the As atoms[37]. Formation of the intra-unit-cell charge/orbital density wave affects the Fe–As hybridization and modifies the hyperfine coupling via isotropic Fermi-contact and core-polarization interactions, accounting for the nearly isotropic $^{75}\mathcal{K}$ Knight shift (Fig. 2). One may note that due to dipole or orbital hyperfine interactions, the Knight shift can split for a field parallel to (110), because half of the As sites have paramagnetic neighbors along (110), and half along (1$\bar{1}$0). However, the difference in the $d$ orbital occupations between nonmagnetic and paramagnetic Fe are expected to be small, likely a few percent (see Supplementary Note 5), in which case the splitting will be below detection, consistent with our experiments.

If we assume a nonmagnetic origin, another plausible candidate could be an orbital-selective Mott transition. In this case, the most correlated Fe orbital state, likely $d_{xy}$, experiences a Mott–Hubbard transition, becoming essentially gapped, while the other orbitals remain itinerant. The resulting occupation change in the $d_{xy}$ state of all Fe sites (Fig. 4e) uniformly changes the hyperfine field at the As sites, retaining the $C_4$ symmetry and explaining the nearly isotropic change of $^{75}\mathcal{K}$ (Fig. 2). Indeed a possibility of such transition has been discussed, but, admittedly, not in undoped pnictides, but in more strongly correlated chalcogenides[38] and (strongly underdoped) KFe$_2$As$_2$[39,40]. We note that as compared to the orbital-selective Mott phase, the former vestigial double-Q phase breaks the additional translational symmetry, which may allow experimental distinction by neutron scattering when bigger crystals become available.

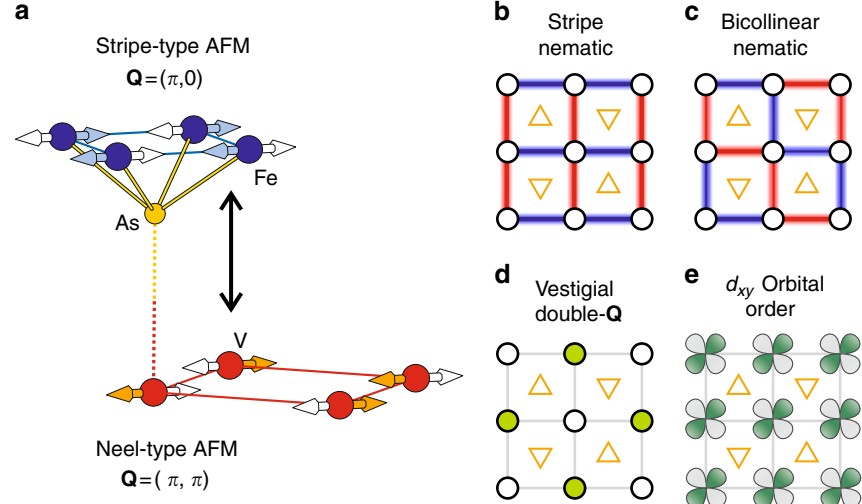

**Fig. 4** Possible orders retaining a $C_4$ symmetry without long-range magnetism. **a** Stripe-type and Neel-type AFM fluctuations of Fe and V spins, respectively, at high temperatures. Two arrows at each Fe and V sites illustrate fluctuating moments with their own spin correlation, indicated by different colors. These different types of AFM fluctuations are frustrated via Fe–V spin coupling (vertical arrow) developed at low temperatures. **b**–**d** Various vestigial ordered phases resulting from melting the corresponding magnetic orders: the typical stripe nematic ($x/y$ symmetry broken), the bicollinear nematic (($x + y$)/($x − y$) and the translation symmetry broken)[36], and the vestigial double-**Q** phase (only the translational symmetry broken)[37]. **e** $d_{xy}$ orbital order driven by a possible orbital-selective Mott transition. As opposed to **b**–**d**, there is no symmetry breaking at all in this phase compared to the high-temperature phase. In **b** and **c**, the symmetry breaks because some bonds are predominantly ferromagnetic (red) and some predominantly antiferromagnetic (blue). In **d**, green circles indicate completely nonmagnetic Fe sites, while open circles correspond to fluctuation paramagnetic sites. The As sites above (up-triangle) and below (down-triangle) the Fe plane are also shown. Note that only **d** and **e** are consistent with the observed $C_4$ symmetry, as discussed in the main text, and are our favorite candidates for the hidden order below $T_0$

As mentioned, $Sr_2VO_3FeAs$ experiences another transition at $T_N \approx 45$ K, which can be identified as a spin density wave highly distinct from the typical stripe AFM. Indeed, $(T_1T)^{-1}$ climbs sharply below 60 K ($\ll T_0$) for both $H \parallel a$ and $H \parallel c$, indicating a critical slow-down of spin fluctuations toward a magnetic ordering at $T_N \sim 45$ K. However, $T_{1,a}^{-1}/T_{1,c}^{-1}$ remains isotropic, suggesting that the coupling between Fe spins and As is due to hybridization, which can only generate a magnetic moment on As if As environment is spin-imbalanced. This excludes such AFM orders as stripe, Neel, or double-**Q**, but would be consistent with a longer period AFM order. Also the progressive broadening of $^{75}$As spectrum at low temperatures, as shown in Fig. 2a, c, suggests a long wavelength, and possibly incommensurate, spin density wave. Neutron diffraction[23,24], which observed magnetic Bragg peaks at $\mathbf{Q} = (1/8, 1/8, 0)$ below $T_N \sim 45$ K, is consistent with this conclusion, although it was incorrectly attributed to an ordering of V spins in the previous studies[16,20–25]. Upon further temperature lowering, $(T_1T)^{-1}$ abruptly drops at $T_c$. This proves that the superconducting gap opens up on the magnetic Fe sites, and emerges on the background of the remaining, but still strong, spin fluctuations with a $C_4$ symmetry below $T_N$. How spin density wave competes or cooperates with superconductivity remains an important question.

**Discussion**

We shall now address an essential question: what suppresses the expected stripe order in the FeAs layer and the Neel order in the $SrVO_3$ layer? The former can be suppressed via the mechanism in which Neel-type spin fluctuations of the localized magnetic moments are coupled to the itinerant electrons' stripe spin fluctuation[12]. The stripe order, with $\mathbf{Q} = (\pi, 0)$ or $(0, \pi)$, is relatively fragile and can give way to bicollinear, double-**Q**, and, possibly, plaquette orders, due to AFM fluctuation with additional **Q**'s[12,37,41]. Such magnetic frustration is due to the long-range

magnetic interactions, reflecting the itinerancy of Fe electrons. Fluctuation at $\mathbf{Q} = (\pi, \pi)$, normally weak in FeSCs, can be enhanced through coupling to the $\mathbf{Q} = (\pi, \pi)$ fluctuations of V spins[12] (Fig. 4a). This destabilizes the $C_2$ stripe AFM or nematic orders, but encourages the $C_4$ symmetric vestigial charge/orbital density wave phases[12,37]. Note that in $Sr_2(Mg,Ti)O_3FeAs$ and $Ca_2AlO_3FeAs$, isostructural compounds with nonmagnetic oxide layers, the standard stripe ordering is not suppressed[42,43]. Clearly additional Neel V spin fluctuations, frustrated with stripe Fe spin fluctuations via magnetic proximity coupling, are essential to stabilize an unusual hidden phase in $Sr_2VO_3FeAs$.

The coupling between the itinerant Fe electrons and the localized V spins also suppress the Neel order in the $SrVO_3$ layer. In the $SrVO_3$ layers, the nearest neighbor superexchange interaction would dominate and generate a stable Neel order. In fact, compared to other $V^{3+}$ perovskite oxides, such as $LaVO_3$, $SrVO_3FeAs$ should have stronger exchange coupling, because of the straighter V–O–V bonds. However, the measured Curie–Weiss temperature of $T_{CW} \sim -100$ K in $Sr_2VO_3FeAs$ is considerably lower than $T_{CW} \sim -700$ K in $LaVO_3$[44]. The unexpectedly low $T_{CW}$ comes from an additional ferromagnetic coupling between the V spins via indirect double-exchange-like interaction mediated by the Fe electrons[45]. This frustrates and weakens the V AFM superexchange interaction suppressing the long-range V spin order at low temperatures. Indeed, in our detailed LDA + U calculations, we found that the calculated magnetic interaction is extremely sensitive to the on-site Coulomb energy $U$ and the Hund's coupling $J$ (note that these corrections were only applied to V, and not to Fe orbitals). At $U - J = 5$ eV, the superexchange interaction, which is inversely proportional to $U$, is significantly suppressed, while the Fe-mediated one is enhanced, so that the net magnetic interaction becomes weakly ferromagnetic in the planes. For $U - J = 4$, it changes sign and becomes antiferromagnetic, consistent with a previous report[16]. This demonstrates that the

SrVO$_3$ lies on the borderline of competing phases due to a delicate balance between the superexchange and the additional indirect interactions. At the same time, coupling between the stripe fluctuations in the Fe plane at $\mathbf{Q} = (\pi, 0)$ and Neel fluctuations in the V plane $\mathbf{Q} = (\pi, \pi)$ suppresses both orders even further[12] and prevents V spins form ordering. The interfacial Fe–V interaction is again crucial for the Mott-insulating SrVO$_3$ layers to remain in a nearly paramagnetic ground state. Our findings therefore manifest that the physics of FeSCs can become even richer in the proximity of other correlated systems and also offer an avenue for exploring unusual ground state in the correlated heterostructures.

## Methods

**Crystal growth.** Single crystals of Sr$_2$VO$_3$FeAs were grown using self-flux techniques as follows. The mixture of SrO, VO$_3$, Fe, SrAs, and FeAs powders with a stoichiometry of Sr$_2$VO$_3$FeAs:FeAs = 1:2 were pressed into a pellet and sealed in an evacuated quartz tube under Ar atmosphere. The samples were heated to 1180 °C, held at this temperature for 80 h, cooled slowly first to 950 °C at a rate of 2 °C/h and then furnace-cooled. The plate-shaped single crystals were mechanically extracted from the flux. High crystallinity and stoichiometry are confirmed by the XRD and energy-dispersive spectroscopy. The typical size of the single crystals is $200 \times 200 \times 10\ \mu m^3$.

**Single crystal characterization.** Single-crystal XRD patterns were taken using an STOE single crystal diffractometer with image plate. Single crystal XRD reveals a good crystallinity in a tetragonal structure with $a = 3.9155(7)$ Å and $c = 15.608(4)$ Å, consistent with the previous studies on polycrystalline samples. Detailed information about single crystal XRD can be found in the Supplementary Information.

Conventional four-probe resistance of single crystals was measured in a 14 T Physical Property Measurement System. Single-crystal magnetizations were measured in a 5 T Magnetic Property Measurement System. The size of one crystal was too small (~0.15 mg) to measure the magnetization, thus 8 pieces of Sr$_2$VO$_3$FeAs single crystals (1.2 mg) were stacked together. All single crystals were carefully aligned along the $c$-axis or the $ab$-plane.

These measurements further confirm the quality of our single crystals. In heterostructured compounds, antisite mixing, here between Fe and V atoms, is known to be detrimental to maintain their intrinsic properties. The As NMR line width and the superconducting transition width are particularly sensitive to the antisite mixing. In our crystals, we found that the As NMR line width is ~30 kHz at 280 K, comparable with typical values of ~5–40 kHz found in other single crystalline FeSCs. Also that the V NMR line width of our single crystal is ~50 kHz at 280 K, much smaller than ~160 kHz, observed in polycrystalline sample[26]. Furthermore we also found that the superconducting transition in our Sr$_2$VO$_3$FeAs single crystal has a sharper resistive transition with a temperature width of $\Delta T_c \sim 3$ K, than found in polycrystalline Sr$_2$VO$_3$FeAs whose $\Delta T_c$ is typically larger than 5 K[13]. These observations consistently suggest that the antisite mixing, if any, cannot be sufficient to induce the observed unusual behavior in our Sr$_2$VO$_3$FeAs single crystal.

**Nuclear magnetic resonance.** $^{51}$V (nuclear spin $I = 7/2$) NMR and $^{75}$As ($I = 3/2$) NMR measurements were carried out at external magnetic fields of 14.983 and 11.982 T, respectively. The sample was rotated using a goniometer for the exact alignment along the external field. The NMR spectra were acquired by a standard spin–echo technique with a typical $\pi/2$ pulse length 2–3 μs and the spin–lattice relaxation rate was obtained by a saturation method.

**Band structure calculations.** Band structure calculations were performed using two standard codes: an all-electron linearized augmented plane wave method implemented in the WIEN2k package[46], and a pseudopotential VASP code[47]. In both cases the gradient-corrected functional of Perdew, Burke, and Ernzerhof was used, and special care was taken to ensure proper occupancy of V orbitals in the LDA + $U$ calculations. LDA + $U$ calculations are known to occasionally converge to metastable minima with incorrect orbital occupancy. Some calculations in the literature suffer from this problem. We ensured, by a proper selection of the starting configuration, that our calculations converge to the correct occupancy, and verified that by analyzing the calculated density of states for each independent run.

**Data availability.** All relevant data are available from the authors.

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

## Acknowledgements

The authors thank L. Boeri, R. Fernandes, S. Backes, R. Valenti, C. Kim, Y. K. Kim, J.-H. Lee, and K. H. Kim for fruitful discussion. This work was supported by the NRF through SRC (No. 2011-0030785) and MPK (No. 2016K1A4A4A01922028). The work at IFW Dresden has been supported by the Deutsche Forschungsgemeinschaft (Germany) via DFG Research Grants BA 4927/1-3 and the Priority Program SPP 1458. Financial support through the DFG Research Training Group GRK1621 is gratefully acknowledged. E.G.M was supported by the POSCO Science Fellowship of POSCO TJ Park Foundation and also NRF (No. 2017R1C1B2009176). I.M. acknowledges funding from the Office of Naval Research (ONR) through the Naval Research Laboratory's Basic Research Program, and from the A. von Humboldt Foundation.

## Author contributions

J.S.K., J.M.O., and S.H.B. conceived the experiments. J.M.O. synthesized the samples. J.M.O. carried out the transport and magnetization measurements. S.H.B. and B.B. contributed to the NMR measurements and the analysis. C.H., R.K.K., S.Y.P., S.D.J., and J.H.P. contribute to single-crystal X-ray diffraction measurements. I.M., S.I.H., J.H.S., Y.B., and E.G.M. contributed to the theoretical calculations and the analysis. J.M.O., S.H.B., E.G.M., I.M., and J.S.K. co-wrote the manuscript. All authors discussed the results and commented on the paper.

## Additional information

**Competing interests:** The authors declare no competing financial interests.

