## [Peer Review File · Nature Communications]

Reviewers' comments:

Reviewer #1 (Remarks to the Author):

The manuscript by Jong Mok Ok and coauthors reported ^{75}As and ^{51}V NMR investigations on high quality single crystal of $\text{Sr}_2\text{VO}_3\text{FeAs}$, a naturally-assembled heterostructure of a Fe-based superconductor and a Mott-insulating vanadium oxide. The authors found a dramatic shift of the ^{75}As NMR line to high frequency below 155 K, while the ^{51}V line shows no significant shift down to 20 K. The $1/T_1T$ of ^{75}As shows a peak at about 200 K and then decreases to a constant below 155K. The results suggest the transition at 155 K occurs in the FeAs layers, not the SrVO_3 layers, without breaking time reversal symmetry and the underlying tetragonal lattice symmetry. Based on their results, the authors proposed the state below 155K to be a C_4 -symmetric charge/orbital order. Moreover, the authors owe the absence of stripe order in the FeAs layers and the Neel order in the SrVO_3 layers to the interactions between the itinerant Fe electrons and the localized V spins.

$\text{Sr}_2\text{VO}_3\text{FeAs}$ experiences a second-order phase transition at about 155K with no evidence of long-range magnetic order. The nature of this transition is still not understood. The authors found new and interesting properties of the state after the transition, and proposed it to be a C_4 -symmetric charge/orbital order which has not been observed in other FeSCs. The results are new and the analysis are reasonable. I recommend to publish this work in Nature Communication.

Reviewer #2 (Remarks to the Author):

In general this is a very interesting paper on NMR results from single crystals of the $\text{Sr}_2\text{VO}_3\text{FeAs}$ (21311) iron-based superconductor. Since the initial report of superconductivity at 37 K in 2009 (reference 12), the 21311 phase has been of great interest since the crystal structure can be viewed as forming a natural superlattice of a Mott insulator and an iron-based superconductor. The 21311 phase, however, is difficult to synthesize, and is susceptible to anti-site defects (mixing of Fe and V), which has resulted in limited work on this material. The availability of small single crystals such as prepared by the authors of the present manuscript or reported in the ARPES work of reference 15 have greatly helped our understanding of this interesting material.

The NMR results are striking and somewhat surprising, and in my opinion justify the publication of this work in Nature Communications, after the authors have addressed the following points:

1. In the abstract the term "high quality crystals" does not convey any useful information. A term like well-characterized crystals would be more accurate. Given the tendency of the 21311 phase to mix V into the Fe sites, the authors should provide some limits from their single crystal refinements as to what extent this mixing is present or absent in their crystals. It may be possible to extract this type of information from their NMR results since a significant amount of mixing would presumably create different local As environments. In polycrystalline sample the superconducting transition width as measured by resistivity measurements is rather broad. Is it any sharper in the single crystals?
2. Since I am not an NMR expert, I found the cartoon illustration (Fig 5 of Suppl. Info.) of the behavior of the Knight shift and spin lattice relaxation rate in the various classes of iron-based superconductors very useful, since one can see at a glance that the phase transition at 155 K in the 21311 phase is unique. I would suggest moving this figure into the main body of the article. Also in the figure caption (Fig 5), the authors have the labels for a and b reversed. Cartoon a should be for FeSe.
3. The spin-density wave ordering in the Fe layer at 45 K followed by superconductivity in the same layer at 37 K is unusual and either points to chemical inhomogeneity in the crystal, or ground states that are extremely close in energy. The authors might want to mention that

something similar happens in FeSe single crystals at pressures near 6 GPa. See for example Fig 1 in Sun et al. Nat Commun. 7:12146 pub July 2016.

4. The authors propose several types if possible orderings for the 155 K transition that are consistent with the NMR and crystallographic data. Are there any experiments that can distinguish between the various possibilities?

Reviewer #3 (Remarks to the Author):

The manuscript reports a study of the ordered phase of Sr₂VO₃FeAs, a compound which can be seen as a natural heterostructure formed by the FeAs layers characteristic of iron-based superconductors and by layer of SrVO₃, a paradigmatic Mott insulator. The material undergoes a transition at 155 K (T₀) which is the main topic of the present work. Using NMR on both ⁷⁵As and ⁵¹V nuclei and a theoretical support based on density-functional calculations including electronic interactions within a DFT+U scheme, the authors show that the transition at T₀ involves the FeAs layers and they rule out the phases which typically occur in other iron-based superconductors and related compounds (stripe, double stripe and structural transition) and they suggest that the experimental results are instead compatible with a C₄ conserving charge/orbital ordering .

This is an interesting paper, which shows a a subtle and non-trivial mutual effect between FeAs layers and vanadium oxide, which leads to the disappearance of the magnetic phases characteristic of the two isolated compounds. This proves that the energetic balance behind, e.g., the stripe phase is very delicate and it can be rather easily modified. For these reasons, I think that this manuscript can be seriously considered for publication in Nature Communications and I have only a few comments that I would like to be addressed.

1) In my experience, it is not so common that heterostructuring with a 1:1 pattern induces such strong modifications to the magnetic properties. I think that some more comments about the specific properties of the present system (or instead some comments about the generality of the results) would be important. In other words, is this result intrinsically related to the elusive properties of iron-based superconductors?

2) Towards the end of page 8 I would like to read some more details on the reason why the coupling to (π,π) fluctuations destabilises the stripes and favour C₄ symmetric solutions. In particular, is these intrinsically related to fluctuating magnetism or a well-ordered phase with little fluctuations would have the same effect?

3) In page 9 the authors connect the reduced temperature for a Curie-Weiss behaviour in the present material with a ferromagnetic coupling competing with the AFM coupling. I am not sure I agree with this argument. The onset of a Curie-Weiss behaviour is indeed associated with the formation of local magnetic moments, which are the effect of Hubbard-like interactions and they are not so much affected by the size of the couplings. Can the authors elaborate more on this point?

4) The value of U-J of 5eV seems significantly larger than any estimate for iron-based superconductors. Can the authors comment about this? Do they expect that the approximate DFT+U treatment requires a larger value of U-J to obtain the correct physics with respect to a more accurate calculation? Finally, the authors talk about a sensitivity of the results on U and J, but in the present calculation the two interactions only enter in the combination U-J. This should be mentioned in the main text.

5) The methods part is extremely short and incomplete and more details should be given to make the results reproducible. In the theory part I read a sentence which definitely requires some clarification: "special care was taken to ensure proper occupancy of V orbitals". How is this realised

in practice? Simply varying the value of the interactions or what?

I think that the manuscript could be published in Nature Communications once all these points are seriously taken into account.

Response to Referees

Reply to Reviewer #1

Q1. The authors found new and interesting properties of the state after the transition, and proposed it to be a C4-symmetric charge/orbital order which has not been observed in other FeSCs. The results are new and the analysis are reasonable. I recommend to publish this work in Nature Communication.

A1. We appreciate Reviewer #1 for his/her positive consideration.

Reply to Reviewer #2

Q1. In the abstract the term “high quality crystals” does not convey any useful information. A term like well-characterized crystals would be more accurate. Given the tendency of the 21311 phase to mix V into the Fe sites, the authors should provide some limits from their single crystal refinements as to what extent this mixing is present or absent in their crystals. It may be possible to extract this type of information from their NMR results since a significant amount of mixing would presumably create different local As environments. In polycrystalline sample the superconducting transition width as measured by resistivity measurements is rather broad. Is it any sharper in the single crystals?

A1. Following the reviewer’s suggestion, we have replaced the term “high quality single crystals” by “single crystals”. Since our work reports the first NMR study on Sr₂VO₃FeAs single crystal, this change does not affect the novelty of our work.

As the reviewer mentioned, the single crystal quality, particularly regarding the anti-site mixing, can be characterized by the NMR spectroscopy since it serves as a local and element-specific probe. The line width of the As NMR spectrum, usually taken near the room temperature well above the magnetic transitions, is known to be sensitive to the non-uniform local environment in iron-based superconductors (FeSCs). Unfortunately, in Sr₂VO₃FeAs the room temperature is rather close to the magnetic instability near 200 K, at which 1/T₁T of the As NMR exhibits a broad maximum. Thus, the As NMR line width is partly determined by magnetic broadening, which hampers its direct comparison with other FeSCs. Nevertheless, the As NMR line width of ~ 30 kHz, observed in our Sr₂VO₃FeAs single crystal at 280 K, is comparable with the typical values of ~ 5 kHz – 40 kHz found in other single crystalline FeSCs. We note that in a BaFe₂As₂ single crystal grown in Sn flux, 1.5% of Sn substitution results in the As NMR line width as large as ~150 kHz [K. Kitagawa et al. J. Phys. Soc. Jpn. 77, 114709 (2008)], much larger than in our case. Although we cannot rule out the anti-site mixing entirely, we believe that it is too weak to affect the intrinsic properties of Sr₂VO₃FeAs. Note also that the V NMR line width of our single crystal is ~ 50 kHz at 280 K, much smaller than ~ 160 kHz, observed in polycrystalline samples [K. Ueshima *et al.* Phys. Rev. B 89, 184506 (2014).].

Furthermore, we also found that the superconducting transition in our Sr₂VO₃FeAs single crystal has a sharper resistive signature, with the width of $\Delta T_c \sim 3$ K, than that in polycrystalline Sr₂VO₃FeAs, where ΔT_c is, typically, larger than 5 K [X. Zhu *et al.* Phys. Rev.

B 79, 220512 (2009).]. These observations consistently suggest that the anti-site mixing, if any, is not at all significant and cannot induce the observed unusual behavior in our $\text{Sr}_2\text{VO}_3\text{FeAs}$ single crystal.

In the revised manuscript, we added one paragraph describing the sample quality, based on NMR line width and also the resistive transition width, in the *Method* section.

Q2. Since I am not an NMR expert, I found the cartoon illustration (Fig 5 of Suppl. Info.) of the behavior of the Knight shift and spin lattice relaxation rate in the various classes of iron-based superconductors very useful, since one can see at a glance that the phase transition at 155 K in the 21311 phase is unique. I would suggest moving this figure into the main body of the article. Also in the figure caption (Fig 5), the authors have the labels for a and b reversed. Cartoon a should be for FeSe.

A2. We appreciate the reviewer's suggestion. We found however that including Fig. S5 in the main text is not possible because of the space restrictions. Instead, we emphasize this comparison more strongly in the revised manuscript, in order to guide the readers toward the supplementary Fig. S5.

The figure caption in Fig S5 is corrected in the revised supplementary information.

Q3. The spin-density wave ordering in the Fe layer at 45 K followed by superconductivity in the same layer at 37 K is unusual and either points to chemical inhomogeneity in the crystal, or ground states that are extremely close in energy. The authors might want to mention that something similar happens in FeSe single crystals at pressures near 6 GPa. See for example Fig 1 in Sun et al. Nat Commun. 7:12146 pub July 2016.

A3. First of all, we would like to emphasize that our NMR results unambiguously confirm the microscopic coexistence of the spin-density-wave (SDW) and superconducting (SC) orders. As shown in Fig. 3, the temperature dependence of $1/T_1T$ exhibits both anomalies due to SDW and SC ordering together. Similar results have been observed in other FeSCs, particularly in the underdoped regime. For examples, As NMR studies on BaFe_2As_2 single crystals, doped with P at the As site, Ru at the Fe site, or K at the Ba site, provided the evidence of microscopic coexistence of these orders, rather than phase separation due to chemical inhomogeneity [T. Iye et al. J. Phys. Soc. Jpn. 81, 033701 (2012), L. Ma et al, Phys. Rev. Lett. 109, 197002 (2012), Z. Li et al, Phys. Rev. B 86, 180501 (2012)].

As the reviewer mentioned, we believe this is an important issue which certainly deserves further study. However, this is beyond the scope of the current work, and would be better covered in the separate study.

Q4. The authors propose several types if possible orderings for the 155 K transition that are consistent with the NMR and crystallographic data. Are there any experiments that can distinguish between the various possibilities?

A4. Since both alternative scenarios break the same rotational symmetries, it is very hard to

distinguish the two by conventional methods. However, one of the phases, the vestigial double-Q phase, breaks (lowers) the translational symmetry in addition. When bigger single crystals will become available, it should be possible to tell them apart by neutron scattering. In the revised manuscript, we mention possible experiments for determining the C4 order at T_0 , as the reviewer suggested.

Reply to Reviewer #3

Q1. In my experience, it is not so common that heterostructuring with a 1:1 pattern induces such strong modifications to the magnetic properties. I think that some more comments about the specific properties of the present system (or instead some comments about the generality of the results) would be important. In other words, is this result intrinsically related to the elusive properties of iron-based superconductors?

A1. We completely agree that such a strong modification of the ground state is not common. As emphasized in the introduction, $\text{Sr}_2\text{VO}_3\text{FeAs}$, consisting of the iron-based superconductor (FeSC) and the Mott insulator, is a new heterostructure system, which may open a novel avenue to explore unprecedented phases and properties of FeSCs.

We believe that the strong effect of the magnetic proximity coupling comes from the itinerant magnetism in FeSCs, combined with the strong-correlations behavior in V oxides. Unlike insulating magnets, in which exchange interactions between local moments determine their magnetic ground state, metallic magnets with conduction electrons forming the magnetic moment often have highly degenerate magnetic ground states, related to the Fermi surface instabilities. For FeSCs, besides the most-commonly observed stripe AFM phase with a magnetic ordering vector of $\mathbf{Q} = (\pi, 0)$, other competing magnetic ground states have been observed and proposed. Frustration of magnetic interactions with different \mathbf{Q} 's can significantly suppress the long-range magnetic order, but not the corresponding nematic (vestigial) orders. The most dramatic example is FeSe, in which the C_2 nematic order is clearly observed with a fully suppressed long-range magnetic order [J. K. Glasbrenner et al. Nat. Phys. (2015)]. The situation is even more intriguing in the case of $\text{Sr}_2\text{VO}_3\text{FeAs}$, because even C_2 nematic phase is fully suppressed, leading to unusual C_4 charge/orbital ordered phase, triggered by the additional interfacial magnetic interaction with $\mathbf{Q} = (\pi, \pi)$. Therefore, the observation of C_4 symmetric order clearly demonstrates that the balance between different magnetic ground state in FeSCs is more subtle than may have been anticipated. Furthermore this newly-discovered C_4 order can microscopically coexist with superconductivity. Whether the observed C_4 order compete or collaborate with the superconducting order remains a challenging question that may lead to deeper understanding of the superconductivity in FeSCs.

Q2. Towards the end of page 8 I would like to read some more details on the reason why the coupling to (π, π) fluctuations destabilises the stripes and favour C_4 symmetric solutions. In particular, is these intrinsically related to fluctuating magnetism or a well-ordered phase with little fluctuations would have the same effect?

A2. According to theoretical predictions [X. Wang et al, Phys. Rev. B 89, 144502 (2014) *ibid.*

91, 024401 (2015); R. M. Fernandes et al. Phys. Rev B 93, 014511 (2016)], the magnetic ground state is determined by two magnetic order parameters, \mathbf{M}_1 and \mathbf{M}_2 , associated with two ordering vectors $\mathbf{Q}_1=(\pi,0)$ and $\mathbf{Q}_2=(0,\pi)$. Then, the nature of the magnetic ground state is defined by the relative magnitude of quartic terms, $(\mathbf{M}_1^2+\mathbf{M}_2^2)^2$, $(\mathbf{M}_1^2-\mathbf{M}_2^2)^2$, and $(\mathbf{M}_1\cdot\mathbf{M}_2)^2$. In an itinerant magnetic system, like FeSCs, their coefficients can be of either sign, which can stabilize C_4 orders rather than the C_2 stripe order. These coefficients, usually set by the relative size or the ellipticity of the electron pockets, are significantly affected by the additional Neel fluctuations, which enhance $\mathbf{Q}=(\pi,\pi)$ interaction and thus induce a C_4 order. Regarding the specific question by the referee, the smaller is the energy difference between the C_2 stripe phase and the C_4 noncollinear or non-uniform phases (in other words, the weaker the biquadratic coupling), the less fluctuations are needed to convert the C_2 phase into the C_4 one. Given the robust indications of strong stripe-type correlation above 200 K in our experiment [Fig. 3], it is reasonable to assume a sizable biquadratic coupling, i.e. a sizable $(\mathbf{M}_1^2-\mathbf{M}_2^2)^2$, so fluctuations of considerable strength are needed for this scenario.

In our revised manuscript, however, we do not include the detailed discussion above and refer the readers to the previous studies. This is because the theoretical description mentioned above is not our results but a prediction from R. Fernandes group, and it should be considered as a plausible scenario rather than a concrete explanation. We would like to emphasize more the experimental finding of a novel C_4 symmetric phase, without being biased, and to stimulate further theoretical studies. Nevertheless, in order to address this referee's specific question, we added in the revised manuscript one sentence emphasizing the importance of the Neel fluctuations.

Q3. In page 9 the authors connect the reduced temperature for a Curie-Weiss behaviour in the present material with a ferromagnetic coupling competing with the AFM coupling. I am not sure I agree with this argument. The onset of a Curie-Weiss behaviour is indeed associated with the formation of local magnetic moments, which are the effect of Hubbard-like interactions and they are not so much affected by the size of the couplings. Can the authors elaborate more on this point?

A3. We would like to note that the CW moments are not much affected by the size of the coupling, as the reviewer mentioned, but the CW temperature is defined entirely by the intersite coupling. A negative CW temperature is a well-known empirical measure of the strength of AFM interactions between the local moments. Experimentally, we found that $\text{Sr}_2\text{VO}_3\text{FeAs}$ has a much weaker coupling ($T_{\text{CW}} \sim -100$ K) than typical vanadium oxides with the same V oxidation state (for example, $T_{\text{CW}} \sim -700$ K of LaVO_3). Such a large reduction of the AFM interaction between V spins is difficult to understand if the vanadium oxide layers are considered in isolation. Thus, we conclude that the coupling to the metallic FeAs layer provides a ferromagnetic interaction between the V-moments (through the metallic double-exchange mechanism), which reduces the net AFM coupling strength.

Q4. The value of U-J of 5eV seems significantly larger than any estimate for iron-based superconductors. Can the authors comment about this? Do they expect that the approximate DFT+U treatment requires a larger value of U-J to obtain the correct physics with respect to a more accurate calculation? Finally, the authors talk about a sensitivity of the results on U and J, but in the present calculation the two interactions only enter in the combination U-J.

This should be mentioned in the main text.

A4. We emphasize that we did not apply any Hubbard U correction to Fe orbitals at all, but only to V d orbitals. $U-J = 4.5-5$ eV is a typical value for correlated V oxides, see, for instance, [M. De-Raychaudhury, E. Pavarini, O.K. Andersen, Phys. Rev. Lett. 99, 126402 (2007)]

In order to avoid unnecessary confusion, we emphasize that Hubbard U correction is applied to V d orbitals not Fe orbitals in the revised manuscript

Q5. The methods part is extremely short and incomplete and more details should be given to make the results reproducible. In the theory part I read a sentence which definitely requires some clarification: "special care was taken to ensure proper occupancy of V orbitals". How is this realised in practice? Simply varying the value of the interactions or what?

A5. Following the referee's suggestion, we provided more detailed information in the *Method* section in the revised manuscript, which had been previously relegated to the Supplementary information. In particular, for the theory part, the following text is added to the *Methods* section.

"LDA+U calculations are known to occasionally converge to metastable minima with incorrect orbital occupancy. Some calculations in the literature suffer from this problem. We ensured, by a proper selection of the starting configuration, that our calculations converge to the correct occupancy, and verified that by analyzing the calculated density of states for each independent run"

Summary of Changes

1. Following the reviewer #2's suggestion, we change the term "high quality single crystals" to "single crystals" in the revised manuscript. [Q1 of Reviewer #2]

[Abstract, 5th sentence]

Here, using high-quality single crystals and high-accuracy ^{75}As and ^{51}V nuclear magnetic resonance (NMR) measurements, we show ...

→ Here, using high-accuracy ^{75}As and ^{51}V nuclear magnetic resonance (NMR) measurements on single crystals, we show...

[Paragraph 3, 2nd sentence]

Using high-accuracy ^{75}As and ^{51}V NMR measurements on high-quality single crystals under various field orientations, ...

→ Using high-accuracy ^{75}As and ^{51}V NMR measurements on single crystals under various field orientations, ...

[Paragraph 4, 1st sentence]

Our transport and magnetic measurements on high-quality single crystal of $\text{Sr}_2\text{VO}_3\text{FeAs}$ shown in Figs. 1b and 1c...

→ Our transport and magnetic measurements on single a crystal of $\text{Sr}_2\text{VO}_3\text{FeAs}$ shown in Figs. 1b and 1c...

2. In the revised manuscript, we added one sentence to emphasize that the NMR properties of $\text{Sr}_2\text{VO}_3\text{FeAs}$ are distinctly different from other FeSCS, as reviewer #2 suggested. [Q2 of Reviewer #2].

[Paragraph 5, 5th sentence]

We emphasize that these behaviors of ^{75}As NMR have never been observed so far in other FeSCs as clearly shown in Supplementary Fig. S5.

3. Following the Reviewer #2's suggestion, we added one sentence to suggest possible experiments to distinguish between two possible phases in the revised manuscript. [Q3 of Reviewer #2]

[Paragraph 10, the last sentence]

We note that as compared to the orbital-selective Mott phase, the former vestigial double-Q phase breaks the additional translational symmetry, which may allow experimental distinction by neutron scattering when bigger crystals become available.

4. We emphasize the importance of the V Neel spin fluctuations by adding one sentence as Reviewer #3 suggested. [Q2 of Reviewer #3]

[Paragraph 12, the last sentence]

Clearly additional Neel V spin fluctuations, frustrated with stripe Fe spin fluctuations via

magnetic proximity coupling, are essential to stabilize an unusual hidden phase in Sr₂VO₃FeAs.

5. In the revised manuscript, we clearly stated that the Hubbard U correction was applied to V orbitals not Fe orbitals [Q4 of Reviewer #3]

[Paragraph 13, 7th sentence]

Indeed, in our detailed LDA+U calculations we found that the calculated magnetic interaction is extremely sensitive to the on-site Coulomb energy U and the Hund's coupling J.

→ Indeed, in our detailed LDA+U calculations we found that the calculated magnetic interaction is extremely sensitive to the on-site Coulomb energy U and the Hund's coupling J (not that these correction were only applied to V, and not to Fe orbitals).

6. In the Method section of the revised manuscript, we added one paragraph with a detailed discussion of the crystal quality based on the NMR line width and the superconducting transition width, as Reviewer #2 suggested. [Q1 of Reviewer #2]

[Method, Paragraph 2]

These measurements further confirm the quality of our single crystals. In heterostructured compounds, antisite mixing, here between Fe and V atoms, is known to be ... the anti-site mixing, if any, cannot be sufficient to induce the observed unusual behavior in our Sr₂VO₃FeAs single crystal.

7. We also added one paragraph describing the details of the band structure calculations, in the Method section of the revised manuscript, as Reviewer #3 suggested. [Q5 of Reviewer #3]

[Method, Paragraph 6]

LDA+U calculations are known to occasionally converge to metastable minima ... verified that by analyzing the calculated density of states for each independent run.

8. The figure caption in Fig S5 was corrected in the revised supplementary information, following the Reviewer #2's suggestions.

[FIG S5, caption]

a The C₂ orbital-ordered and AFM phases, the most commonly observed parent phase in AFe₂As₂, ReOFeAs, and A₂MO₃FeAs (A = alkaline earths, Re = rare earths, M = nonmagnetic metals). **b** The C₂ orbital ordered phase without any static AFM order in FeSe.

→ **a** The C₂ orbital ordered phase without any static AFM order in FeSe. **b** The C₂ orbital-ordered and AFM phases, the most commonly observed parent phase in AFe₂As₂, ReOFeAs, and A₂MO₃FeAs (A = alkaline earths, Re = rare earths, M = nonmagnetic metals).

REVIEWERS' COMMENTS:

Reviewer #2 (Remarks to the Author):

The authors have addressed all of my questions and concerns and I believe the paper is now suitable for publication in Nature Communications.

Reviewer #3 (Remarks to the Author):

The authors revised the manuscript after the reports by myself and two other referees. I find the individual responses quite satisfactory overall, even if sometimes the changes to the manuscript are limited. However, I think that the revised manuscript overcomes almost all the problems of the original version and it can be recommended for publication.

The only point I would like the authors to address is to add some very short comment about by original first question

Q1. In my experience, it is not so common that heterostructuring with a 1:1 pattern induces such strong modifications to the magnetic properties. I think that some more comments about the specific properties of the present system (or instead some comments about the generality of the results) would be important. In other words, is this result intrinsically related to the elusive properties of iron-based superconductors?

I think that some of the arguments given by the authors in their reply should be included in the manuscript to better stress one non-trivial aspect of the present analysis.

Response to Referees

Reply to Reviewer #3

Q1. *The only point I would like the authors to address is to add some very short comment about by original first question*

In my experience, it is not so common that heterostructuring with a 1:1 pattern induces such strong modifications to the magnetic properties. I think that some more comments about the specific properties of the present system (or instead some comments about the generality of the results) would be important. In other words, is this result intrinsically related to the elusive properties of iron-based superconductors?

A1. Following the referee's suggestion, we included the following sentences in the end of introduction. This emphasizes that our surprising findings in the heterostructure based on FeSCs are intimately related to the Fermi surface instabilities in FeSCs, with complex interplay of various degrees of freedom, which is distinct from those found in other heterostructures based on transition metal oxides.

“Such a strong modification of the ground state is not common in other strongly correlated TMO heterostructures [3–5], which highlights that FeSCs, itinerant systems with complex interplay of spin/charge/orbital degrees of freedom, have competing ground states related to the Fermi surface instabilities, and thus are extremely sensitive to additional interfacial interaction in heterostructures.”